# Targeting Lactate Dehydrogenase A with Catechin Resensitizes SNU620/5FU Gastric Cancer Cells to 5-Fluorouracil

**DOI:** 10.3390/ijms22105406

**Published:** 2021-05-20

**Authors:** Jung Ho Han, MinJeong Kim, Hyeon Jin Kim, Se Bok Jang, Sung-Jin Bae, In-Kyu Lee, Dongryeol Ryu, Ki-Tae Ha

**Affiliations:** 1Department of Korean Medical Science, School of Korean Medicine, Pusan National University, Yangsan 50612, Korea; hanjh1013@pusan.ac.kr; 2Healthy Aging Korean Medical Research Center, Pusan National University, Yangsan 50612, Korea; Dr.NowOrNever@pusan.ac.kr; 3Department of Molecular Cell Biology, Sungkyunkwan University School of Medicine, Suwon 16419, Korea; alswjd0105@skku.edu; 4Department of Molecular Biology, College of Natural Science, Busan 46241, Korea; khjkhj0903@naver.com (H.J.K.); sbjang@pusan.ac.kr (S.B.J.); 5Department of Internal Medicine, School of Medicine Kyungpook National University, Daegu 41566, Korea; leei@knu.ac.kr

**Keywords:** 5-fluorouracil, chemoresistant, glycolysis, lactate dehydrogenase, catechin

## Abstract

Resistance to anticancer therapeutics occurs in virtually every type of cancer and becomes a major difficulty in cancer treatment. Although 5-fluorouracil (5FU) is the first-line choice of anticancer therapy for gastric cancer, its effectiveness is limited owing to drug resistance. Recently, altered cancer metabolism, including the Warburg effect, a preference for glycolysis rather than oxidative phosphorylation for energy production, has been accepted as a pivotal mechanism regulating resistance to chemotherapy. Thus, we investigated the detailed mechanism and possible usefulness of antiglycolytic agents in ameliorating 5FU resistance using established gastric cancer cell lines, SNU620 and SNU620/5FU. SNU620/5FU, a gastric cancer cell harboring resistance to 5FU, showed much higher lactate production and expression of glycolysis-related enzymes, such as lactate dehydrogenase A (LDHA), than those of the parent SNU620 cells. To limit glycolysis, we examined catechin and its derivatives, which are known anti-inflammatory and anticancer natural products because epigallocatechin gallate has been previously reported as a suppressor of LDHA expression. Catechin, the simplest compound among them, had the highest inhibitory effect on lactate production and LDHA activity. In addition, the combination of 5FU and catechin showed additional cytotoxicity and induced reactive oxygen species (ROS)-mediated apoptosis in SNU620/5FU cells. Thus, based on these results, we suggest catechin as a candidate for the development of a novel adjuvant drug that reduces chemoresistance to 5FU by restricting LDHA.

## 1. Introduction

Remarkable progress has been achieved in the treatment of gastric cancer with surgical resection of the primary tumor and 5-fluorouracil (5FU)-based combinational chemotherapy, such as epirubicin, cisplatin, and fluorouracil and docetaxel, oxaliplatin, and 5FU [1,2]. To date, 5FU, an analog of uracil that blocks DNA synthesis by inhibiting thymidylate synthase [3], remains a commonly used chemotherapy drug for the treatment of gastric cancer [1,4]. However, the clinical efficacy of chemotherapy has been unsatisfactory owing to the relapse of primary or metastatic cancer related to drug resistance [1,5].

Several cellular processes are regarded as key mechanisms underlying chemoresistance, including drug activation or inactivation, repair machinery of DNA damage, modifications in drug targets, inactivation of apoptotic pathways, and increased autophagy [6,7]. Cancer is heterogeneous and various cancers use both glycolysis and oxidative phosphorylation for energy metabolism [8]. However, metabolic reprogramming, especially abnormal aerobic glycolysis, has been reported as a key player in the development of resistance to chemotherapeutic drugs, including 5FU, in gastric cancer [9]. Lactate dehydrogenase A (LDHA) and pyruvate dehydrogenase kinases (PDKs) are known as key regulators of aerobic glycolysis producing lactate from pyruvate [10,11]. The combination of dichloroacetate, an established inhibitor of PDK, restores sensitivity toward 5FU [12]. In addition, LDHA, a key enzyme catalyzing the conversion of pyruvate to lactate, has been reported as a potential target for the treatment of gastric cancer [13,14]. Knockdown of LDHA expression in cancer cells inhibits tumorigenicity through increasing mitochondrial respiration and reducing cell viability [15]. Several LDHA inhibitors, such as FX11, PSTMB, and GSK 2837808A, diminishes ATP levels and generate oxidative stress, and finally cause apoptotic cell death [16,17,18]. Although a study demonstrated that the inhibition of LDHA by microRNA-34a recovered the resistance to 5FU in colon cancer [19], no previous study has reported the effects of genetic or pharmacological LDHA inhibitors on chemoresistance.

Green tea polyphenols, including catechin (CA), epicatechin (EC), gallocatechin (GC), epigallocatechin (EGC), epicatechin-3-gallate (ECG), and epigallocatechin-3-gallate (EGCG), promote human health by preventing and treating several diseases, including inflammation, diabetes, cardiovascular disease, and malignant cancer [20,21]. Among polyphenols, EGCG enhances 5FU chemosensitivity in colorectal cancer and hepatocellular carcinoma cells by activating AMPK pathways and restricting cancer stemness [22,23]. In addition, EGCG has been reported to inhibit LDHA by repressing its expressions in the breast and pancreatic cancer cells [24,25].

In this study, we hypothesized that CA might be a candidate for an adjuvant drug for reducing the resistance to 5FU in gastric cancer cells. It was examined that the inhibitory effect of CA, and/or its combination with 5FU, on the LDHA activity thereby induced mitochondrial reactive oxygen species (ROS) and apoptotic cell death in 5FU-resistant cells. 

## 2. Results

### 2.1. Glycolytic Characteristics of 5FU-Resistant Cells

Enhanced proliferation is a characteristic of chemoresistance including 5FU-resistant cancer cells [26,27]. First, we confirmed the resistance to 5FU and long-term growth rates of SNU620 and SNU620/5FU cells. When these cells were cultured with the indicated concentrations of 5FU for 48 h, the estimated dose causing 50% growth inhibition (GI_50_) in SNU620/5FU cells (100.63 μM) was much higher than that causing GI_50_ in its parental SNU620 cells (28.88 μM) (Figure 1A and Appendix A). Additionally, the growth rates of SNU620/5FU cells were faster than those of SNU620 cells (Figure 1B and Appendix A). Since chemoresistance and fast-growing characteristics could be related to elevated glycolytic property in gastric cancer cells [28], extracellular acidification rates (ECAR) and oxygen consumption rates (OCR) were measured in both cells in the presence of oxamate, a well-known glycolysis inhibitor, or not. Results demonstrated that SNU620/5FU cells significantly secreted more lactate than that secreted by SNU620 cells and subsequently acidified the culture medium (Figure 1C and Appendix A). However, no significant change in OCR was observed between these cells (Figure 1D). The mRNA and protein expression levels of LDHA were measured. LDHA expression was higher in SNU620/5FU cells than in SNU620 cells (Figure 1E,F). Further, we examined the mRNA and protein expression levels of PDK isotypes and found that those of PDK2 and PDK3 were higher in SNU620/5FU cells than in SNU620 cells (Appendix A). The phosphorylation of the E1α subunit of pyruvate dehydrogenase (PDHA1), which was increased by PDKs, was also evaluated, and it was found to be higher in SNU620/5FU cells than in SNU620 cells (Appendix A). In addition, we used oxamate as an LDHA inhibitor to overcome 5FU resistance in SNU620/5FU cells. This compound inhibited cell growth in both SNU620 and SNU620/5FU cells (Appendix A). Further, cotreatment with oxamate and 5FU suppressed cell growth, compared to individual treatment with oxamate or 5FU in SNU620/5FU cells (Figure 1G). These data suggest that SNU620/5FU cells highly express LDHA, and inhibition of LDHA activity can overcome the resistance of SNU/5FU cells to 5FU.

### 2.2. CA Suppresses LDHA Activity

Lactate production was analyzed with CA and its derivatives, including EC, GC, EGC, and EGCG, in SNU620/5FU cells. Compared with these derivatives, CA reduced lactate production more effectively (Figure 2A). Further, in vitro, the LDHA activity assay was confirmed, and CA showed a stronger inhibitory effect on LDHA activity than those of its derivatives (Figure 2B). The inhibitory effects of CA on in vitro LDHA and LDHB activities were measured in a dose-dependent manner. CA inhibited LDHA activity without affecting LDHB activity (Figure 2C,D). Additionally, LDHA and LDHB activities were measured with EGCG, which is known to inhibit LDHA activity, and were found to have dose-dependently decreased (Appendix A). Moreover, intracellular LDHA and LDHB activities were evaluated with the indicated concentrations of CA in SNU620/5FU cells. CA inhibited LDHA activity in a dose-dependent manner in in vitro results but not LDHB activity (Figure 2E,F). Protein expression levels of p-PDHA1, PDK2, PDK3, and LDHA were also measured in SNU620/5FU cells after CA treatment (Appendix A), and we observed that those were not affected by this treatment. These results suggest that CA regulates LDHA activity without affecting its expression.

### 2.3. Mode of Action of LDHA Inhibition by CA

To analyze the inhibition mechanism of the LDHA activity on CA and its derivatives, we conducted the molecular modeling of the binding interaction between human recombinant LDHA and CA (Figure 3A,B). The results showed that CA may interact with the residues of T94, A95, Q99, R105, S136, R168, H192, and T247 of LDHA. Among them, R105, S136, and R168 are critical for substrate (pyruvate) binding. Thus, we assumed that CA might interfere with the substrate-binding sites. On the contrary, EGCG is bound to LDHA via a different site. To verify the mode of action of LDHA inhibition by CA further, the Michaelis–Menten and Lineweaver–Burk plots were confirmed with various concentrations of pyruvate and CA. LDHA activity was increased in a pyruvate dose-dependent manner. However, it was reversed by the increase of CA. Thus, the results of Michaelis–Menten and Lineweaver–Burk plots showed that the CA is a noncompetitive inhibitor to LDHA (Figure 3C,D). In addition, isothermal titration calorimetry (ITC) data demonstrated that CA physically binds to LDHA by a Kd value of 197 μM (Appendix A). These results provide insights that CA suppresses LDHA activity through binding the substrate-binding site of LDHA.

### 2.4. CA Resensitizes 5FU Resistance through Reducing LDHA Activity

We evaluated the cytotoxic effects of CA on SNU620 and SNU620/5FU cells and found that CA suppressed growth in both groups of cells (Figure 4A). To verify that the growth inhibition was related to the suppression of LDHA activity, the expression of LDHA was abolished by short hairpin ribonucleic acid (shRNA) for LDHA (shLDHA) and its expression was recovered by transfection LDHA expressing plasmid (Appendix A). In shLDHA-transfected SNU620/5FU cells, the lactate production and cell viability were not markedly changed by CA treatment, compared to these of mock-transfected cells (Figure 4B,C). In addition, the growth inhibitory effect of CA was increased by the recovery of LDHA expression in LDHA-knockdown SNU620/5FU cells (Figure 4D). Further, SNU620 and SNU620/5FU cells were examined following the treatment of CA and 5FU individually or the combination of both. Cotreatment with CA and 5FU significantly inhibited the growth of SNU620/5FU cells (Figure 4E). Additionally, combination effects of CA and 5FU were examined in several other cell lines, such as human gastric cancer AGS, pancreatic cancer Panc-1, and MIA PaCa-2, and colon cancer LS174T and RKO cells, compared to SNU620 and SNU620/5FU cells. The lactate production and LDHA expression showed a good correlation with each other, and these glycolytic properties are relatively high in SNU620/5FU, AGS, and RKO (Appendix A). Moreover, the combination treatment of CA and 5FU significantly reduced the viability of cells, compared to a single treatment of 5FU in glycolytic cells including AGS and RKO (Appendix A). However, cotreatment with CA and 5FU did not significantly inhibit growth in SNU620, Panc-1, MIA PaCa-2, and LS174T cells, which showed low LDHA expressions. These results suggested that the sensitizing effect of CA to 5FU-resistant cells might be related to its inhibitory effect on LDHA activity.

### 2.5. CA and 5FU Cotreatment Induces Mitochondrial ROS-Dependent Apoptosis

Since LDHA inhibition directly increases the mitochondrial ROS and loss of its membrane potential, subsequently inducing apoptosis [16,29], the factors related to the mitochondrial ROS-mediated apoptosis were examined. CA or 5FU single treatment slightly increased mitochondrial ROS level in SNU620/5FU cells when measured by flow cytometry using MitoSOX. However, cotreatment with CA and 5FU significantly enhanced mitochondrial ROS production. This increased ROS production was reversed by Mito-TEMPO, a mitochondria-targeted antioxidant (Figure 5A–C). Growth inhibition following cotreatment with CA and 5FU was also recovered upon Mito-TEMPO treatment in SNU620/5FU cells (Figure 5D). Further, apoptosis was evaluated via Annexin V–FITC/PI staining in SNU620/5FU cells. Cotreatment with CA and 5FU increased the number of apoptotic cells among SNU620/5FU cells (Figure 6A,B). Nuclear morphologic fragmentation appears during apoptosis, and it can be observed by 4′,6-diamidino-2-phenylindole (DAPI) staining [30,31]. DNA fragmentation was found to have significantly increased in the cotreatment group, compared to that in the control group (Figure 6C). Finally, biomarkers of apoptosis including Bcl-2, BAX, Caspase-9, Caspase-3, and PARP were examined. The proapoptotic cascade increased following cotreatment with CA and 5FU in SNU620/5FU cells (Figure 6D). These findings suggest that cotreatment with CA and 5FU induces apoptosis in SNU620/5FU cells.

## 3. Discussion

In this study, to characterize and evaluate chemoresistance to 5FU, we used SNU620/5FU cells, which were established by long-term exposure to 5FU with a serial increment of drug concentration [32]. The parental SNU620 cells have been reported to harbor a p53 mutant with a homozygous deletion in exon 5 [33]. Based on a DNA microarray, several genes involved in chemoresistance, including thymidylate synthetase, damage-specific DNA binding protein 2, clusterin, and midkine, are elevated in SNU620/5FU cells [32]. Several previous studies have used the SNU620/5FU cell line as a model to evaluate the in vitro efficacy of drugs against resistance to 5FU through anti-mitosis, AMPK activation, and cannabinoid receptor signaling [34,35,36]. However, the metabolic characteristics and expression levels of metabolism-related genes in the SNU620/5FU cells and their roles in chemoresistance have not been largely investigated. 

Here, we demonstrated that SNU620/5FU cells have advanced glycolytic phenotypes, including elevated lactate production and the expression of enzymes related to the conversion of pyruvate to lactate, such as LDHA. The phosphorylation of PDHA1, representing the reduced PDH activity by elevated PDK2 and PDK3, also increased in SNU620/5FU-resistant cells, compared with that in SNU620 parent cells. However, OCR was not significantly decreased in resistant cells. A possible explanation for this discrepancy is the nutrient plasticity of cancer cells for the TCA cycle, that is, amino acids or fatty acid-derived acetyl-CoA, not only glucose-derived acetyl-CoA, could supply substrates to the TCA cycle [37]. In correlation with this, 5FU-resistant gastric cancer cells promote stemness via mitochondrial fatty acid oxidation [38]. In addition, 5FU-resistance has been related to increased mitochondrial mass and activity, including the expression of electron transport chain (ETC) enzymes and oxygen consumption [39,40]. Thus, we focused on the modulation of LDHA to resensitize the 5FU-resistance. Based on our results, the inhibition of LDHA activity with oxamate or CA successfully suppressed the growth of resistant SNU620/5FU cells and resensitized them to 5FU treatment. These findings showed a good correlation with those of previous studies, which reported that genetic or pharmacological inhibition of LDHA successfully reduced resistance to chemotherapy, including 5FU [19,41,42]. Thus, we assumed that the inhibition of LDHA might be sufficient to suppress the resistance against 5FU in SNU620/5FU cells.

Several synthetic molecules, such as oxamate, FX11, PSTMB, and GNE-140 have been established as small-molecule LDHA inhibitors [11,43]. Among these, FX11 has been reported as a sensitizer to chemotherapy in resistant tumor cells [44]. However, although many LDHA inhibitors are under scrutiny for approval as novel anticancer drugs, none of them have been approved yet. Thus, more drug-like candidates are required to generate new LDHA inhibitors [45]. Natural compounds have been regarded as potential resources for anticancer drugs, particularly in overcoming chemoresistance as a combination therapy [46,47]. Therefore, targeting cancer metabolism, especially glucose metabolism with plant-derived natural products is an emerging research trend for the development of novel cancer therapeutics [48].

Further, several reports have suggested that natural products, such as gossypol, galloflavin, crocetin, machilin A, and EGCG, are potent LDHA inhibitors [43,49,50]. Although among these, gossypol has the most potent inhibitory action on LDHA (IC_50_ = 9.8 μM), it also inhibits the activity of LDHB, which converts lactate to pyruvate [51]. Our results demonstrated that EGCG, an established LDHA inhibitor and a sensitizer to 5FU chemotherapy [22,23,24,25], exert inhibitory effects on both LDHA and LDHB activities. Among the natural product-derived LDHA inhibitors showing LDHA-specific inhibitory activities, such as machilin A (IC_50_ = 84 μM), crocetin (IC_50_ = 54.9 μM), and CA (IC_50_ = 40.69 μM) [49,50], CA showed the best suppressive effect on LDHA activity with respect to the IC_50_ value. Although the chemical structures of CA and EGCG are very similar except for the additional gallic acid moiety, their inhibitory selectivities were not the same. In addition, despite CA being the simplest compound, compared with its derivatives, it showed the best inhibitory action on both lactate production and LDHA activity. Thus, an extensive structure–activity relationship study should be conducted to demonstrate CA as a potential novel backbone for the development of specific and potent LDHA inhibitors.

EGCG has been known that a competitive inhibitor of NADPH. In addition, EGCG inhibits the NADPH oxidase translocation and ROS production [52,53,54]. However, according to the molecular modeling results, CA and EGCG are structurally located in different places (Figure 3A, B). In addition, they did not show the correlation in the specificity in the inhibition of LDHA and LDHB activity, which use the NADH and NAD^+^ as a cofactor, respectively. From these data, we assumed that CA has no direct correlation with EGCG in the mechanism of competing with NAD(P)H. By the way, inhibition of LDHA by several inhibitors, such as FX11, GSK 2837808A, NCI-737, and NCI-006, reduced the NAD^+^ production and increased the NADH accumulation, thereby decreasing the ratio of NAD^+^/NADH [16,17,55]. Apoptosis induced by p53/NAD-dependent DNA damage pathway is also raised by LDHA inhibition using siRNA or chemical inhibitor, NHI-2 [56]. The GCN2-ATF4 signaling pathway was also reported as another mechanism responsible for apoptotic cell death induced by LDHA inhibitor, GSK 2837808A [17]. Since the NAD^+^/NADH ratio plays a key role in redox homeostasis and cell proliferation [57,58], the CA as an LDHA inhibitor might affect the 5FU-resistant cancer cells through modulation of the NAD^+^/NADH ratio.

CA induced cell death in both parental SNU620 and 5FU-resistant SNU620/5FU cells. However, CA-induced cell death was LDHA dependent through shRNA depletion in SNU620/5FU cells. In addition, the cotreatment of CA and 5FU showed additional anticancer effects on 5FU-resistant cells. As previously reported [16,59], inhibition of LDHA increases the ROS and consequently suppressed the growth of cancer cells. When LDHA was inhibited, the energy metabolism was converted from glycolysis to the TCA cycle and mitochondrial oxidative phosphorylation [60]. Thus, cancer cells make more ROS production, resulting in the damage of mitochondrial membrane and mitochondrial-induced apoptosis [61,62]. In good agreement with previous studies, CA induces apoptotic cell death through a mitochondrial ROS-dependent pathway. Increased ROS-defense ability and reduced apoptotic signals are common properties of chemoresistant cancer cells, especially in 5FU-resistant cells [3,63]. Further, alterations in cancer metabolism, particularly in glucose metabolism, also lead to resistance to chemotherapy through the alteration of cellular activities, such as aberrant DNA repair, enhanced autophagy, reduced apoptosis, defense against ROS, and increased secretion of exosomes [9,64,65]. Thus, targeting glycolytic enzymes, including LDHA, with CA might be an alternative strategy for overcoming chemoresistance, especially in 5FU-resistant gastric cancer.

Generally, rapidly dividing malignant tumors is highly sensitive to DNA synthesis inhibitors including 5FU, compared to normal tissue. However, some cancer cells can develop resistance to the treatment through several mechanisms, as previously described [34,35,36]. The cotreatment of bioactive compounds and conventional chemotherapy has a higher effect, compared to a single compound, on slowing the development of resistance [66]. Therefore, the biological effects of specific phytochemicals with proven cytotoxic effects administered with conventional chemotherapy to target a wider range of signaling pathways in cancer cells, including cancer metabolism and mitochondrial functions, should be superior to single compounds in cancer management since they may delay the development of resistance [67]. In this study, the combination of CA and 5FU showed a higher inhibition on cell viability, compared to that of a single treatment. Moreover, we demonstrated that the inhibition of LDHA activity and subsequent mitochondrial ROS-mediate apoptosis might be the mechanism underlying the sensitizing effect of CA on 5FU-resistant cells (Figure 7).

In addition to the potency and specificity of LDHA inhibition, CA is safer than previously established natural product-derived LDH inhibitors. Moreover, CA is a well-known chemical ingredient of green tea, and its safety and pharmacodynamic properties have been confirmed in previous studies [68,69,70]. However, a precise toxicity assessment of the coadministration of CA and 5FU has not yet been conducted. In addition, the in vivo efficacy of CA in 5FU-resistant cancer cells was not examined in this study. However, previous studies report on the in vivo anticancer efficacy of CA and its derivatives to overcome chemoresistance, including the 5FU-resistance in human gastric and colon cancer [71,72,73,74,75,76,77]. In this study, we focused on the LDHA inhibition as a major molecular mechanism of CA of overcoming the resistance to 5FU. Therefore, to develop CA as a novel adjuvant for chemoresistant cancer cells, the in vivo efficacy and safety of CA and 5FU cotreatment should be evaluated through extensive animal studies, including xenograft models and good laboratory toxicity assessments.

## 4. Materials and Methods

### 4.1. Materials

Antibodies against poly (ADP-ribose) polymerase, caspase-3, and caspase-9 were purchased from Cell Signaling Technology (Cell Signaling Technology, Danvers, MA, USA). Antibodies against LDHA, glyceraldehyde 3-phosphate dehydrogenase (GAPDH), and PDHA1 were purchased from Santa Cruz Biotechnology (Santa Cruz Biotechnology, Santa Cruz, CA, USA), and those against phosphor-PDHA1 and PDK3 were purchased from Abcam (Abcam, Cambridge, MA, USA). Further, antibodies against PDK2 and PDK4 were purchased from Signalway Antibody (Signalway Antibody, Dallas, TX, USA), those against PDK1 was obtained from Enzo Life Sciences (Enzo Life Sciences, Farmingdale, NY, USA), and those against B-cell lymphoma-2 (Bcl-2) and Bcl-2-associated X protein were purchased from Novus Biologicals (Novus Biologicals, Littleton, CO, USA). MitoSOX was purchased from Invitrogen (Invitrogen, Carlsbad, CA, USA). Chemicals and reagents, including 3-(4,5-dimethylthiazol-2-yl)-2,5-diphenyltetrazolium bromide (MTT), 4′,6-diamidino-2-phenylindole, oxamate, nicotinamide adenine dinucleotide, CA, and CA derivatives (EC, GC, EGC, and EGCG) were purchased from Merck (Merck, Darmstadt, Germany). β-nicotinamide adenine dinucleotide (oxidized form) was purchased from Tokyo Chemical Industry (Tokyo Chemical Industry, Tokyo, Japan).

### 4.2. Cell Culture

Human gastric cancer SNU620, SNU620/5FU, and AGS, pancreatic cancer Panc-1 and MIA PaCa-2, and colon cancer LS174T and RKO cells were obtained from the Korean Cell Line Bank (KCLB, Seoul, Korea). SNU620, SNU620/5FU, AGS, and LS174T cells were cultured in Roswell Park Memorial Institute medium 1640 (RPMI-1640) (Welgene, Daegu, Korea), supplemented with 10% heat-inactivated fetal bovine serum (FBS) (Gibco, New York, NY, USA) and 1% penicillin/streptomycin (Invitrogen), and the Panc-1, MIA PaCa-2, and RKO cells were cultured in Dulbecco’s modified Eagle’s medium (Welgene) containing 10% heat-inactivated FBS and 1% penicillin/streptomycin. All cells were cultured in a humidified CO_2_ incubator at 37 °C and 5% CO_2_.

### 4.3. Cell Viability Assay

The cytotoxicity levels of CA and 5FU in SNU620 and SNU620/5FU cells were measured using MTT assay. Cells were cultured in 24-well plates (2 × 10^4^ cells/well) with the indicated concentrations of CA and 5FU for the stated day. MTT solution (2.0 mg/mL) was then added to each well, followed by 3–4 h of incubation at 37 °C and 5% CO_2_ in a cell culture incubator. The culture medium was subsequently removed, and the absorbance of formazan crystals made from live cells. DMSO was added to dissolve the formazan crystals, and it was measured at 540 nm using a Spectramax M2 Microplate Reader (Molecular Devices, Sunnyvale, CA, USA).

### 4.4. Extracellular Acidification Rate (ECAR) and Oxygen Consumption Rate (OCR)

ECAR and OCR, indicating the cellular rates of glycolysis and oxidative phosphorylation, respectively, were monitored with the Seahorse XF analyzer (Agilent Technologies, Santa Clara, CA, USA), as described previously [78,79]. Briefly, 60,000 SNU620 or SNU620/5FU cells per well were seeded in Seahorse XF six-well plates in RPMI medium supplemented with 1% penicillin/streptomycin. Following a 30 min incubation, the medium from each well was replaced with 80 µL of the prewarmed serum-free medium with 5 mM oxamate, a standard LDH inhibitor [11]. Cells were then incubated at 37 °C for 24 h. After incubation, the medium from each well was replaced with 180 µL of prewarmed XF base medium (containing 10 mM glucose, 2 mM glutamine, and 1 mM sodium pyruvate; pH 7.4) to measure ECAR and OCR. Results were analyzed using the Wave 2.6.0.31 software (Agilent Technologies).

### 4.5. Quantitative Reverse Transcription–Polymerase Chain Reaction (qRT–PCR)

Total RNA was extracted using the RiboEx Total RNA Extraction Kit (GeneAll Biotechnology, Seoul, Korea), and cDNA was synthesized using a reverse transcriptase kit (Promega, Madison, WI, USA). In reverse transcription, 1 μg of total RNA was used, and the total amount of cDNA synthesized is 20 μL. Each kit was used according to the manufacturer’s instructions. Quantitative PCR was performed using a StepOne™ Real-Time PCR System (Thermo Fisher Scientific, Waltham, MA, USA), with the Real Helix qPCR Kit (NanoHelix, Daejeon, Korea), for 40 cycles consisting of 15 s at 95 °C and 1 min at 60 °C. Relative mRNA levels were normalized to the levels of 18S ribosomal RNA, which served as an endogenous control. Sequences of the primers used for qRT–PCR are listed in Table 1.

### 4.6. Western Blot Analysis

The cells were washed with 1× PBS, and total proteins were extracted from cells using RIPA buffer and 1% NP-40 lysis buffer containing protease inhibitor cocktail tablets (Roche, Basel, Switzerland). Each proteins concentrations were measured using the Bio-Rad protein assay. Equal amounts of protein were fractionated from each sample through 8–15% SDS–PAGE, and then the proteins were transferred to nitrocellulose membranes (GE Healthcare, Munich, Germany) via electrophoresis. The membranes were blocked at room temperature (20–25 °C) for 1 h using 5% nonfat dry milk and incubated with primary antibodies at 4 °C overnight. Subsequently, these membranes were washed three times with 1× Tris-buffered saline for 10 min. Specific bands of proteins were measured with a chemiluminescence imaging system (ImageQuant LAS 4000; GE Healthcare). The expression of proteins was adjusted by GAPDH.

### 4.7. Lactate Production Assay

Lactate production was measured in the culture media of SNU620, SNU620/5FU, AGS, Panc-1, MIA PaCa-2, LS174T, and RKO cells. These cells were incubated for 1 d at 37 °C, and the culture media were subsequently replaced with phenol red-free medium, followed by incubation for 1 h at 37 °C. The medium of each cell was then evaluated using a commercial lactate fluorometric assay kit (BioVision, Milpitas, CA, USA).

### 4.8. LDHA and LDHB Activity Assays

To detect LDHA activity, the indicated concentrations of CA were incubated for 20 min in a buffer containing 2 mM pyruvate, 20 μM NADH, and 20 mM HEPES-K^+^ (pH 7.2). For LDHB activity, a buffer containing 1 M Tris-HCl (pH 8.0), 25 mM NAD^+^, and 2 M sodium *L*-lactate was used. Briefly, 10 nanograms of each of the purified recombinant LDHA and LDHB proteins were used for the in vitro LDHA and LDHB activity assays. One microgram of total protein from cell lysates was used for intracellular LDHA and LDHB activity assays as an enzyme source. Fluorescence of NADH at an excitation wavelength of 340 nm and an emission wavelength of 460 nm was detected using a spectrofluorometer (Spectramax M2; Molecular Devices), as previously described [80]. LDHA activity was measured by the decreased amount of NADH, whereas LDHB activity was evaluated by measuring the amount of NADH converted from NAD^+^.

### 4.9. Protein–Small Molecule Interaction

The interaction between protein and small molecules was predicted using the Pyrx program. The LDHA (PDB ID: 1I10) molecules and the 2D structures of CA and EGCG obtained from the NCBI PubChem compound database were used in Pyrx. The ID of CA was 9064 and that of EGCG was 65064. The relative distribution of the surface charge was shown with the acidic region in red, the basic region in blue, and the neutral region in white. Hydrogen bonds in LDHA complexes with CA or EGCG, respectively, were shown as black dotted lines. Sequences were obtained from UniProt (https://www.uniprot.org, accessed on 9 November 2020) with accession numbers P00338 (LDHA).

### 4.10. Transfection of Short Hairpin RNA (shRNA)

The pLKO.1 mock-vector and shRNA targeting LDHA vector were used, as previously described [50]. Cells were seeded at six-well plates (2 × 10^5^ cells/well) and incubated overnight. SNU620/5FU cells were transfected using polyethyleneimine (PEI) (Polyplus-transfection, Illkirch, France), gene and PEI ratio is 1:3. Then, transfected cells were treated with 1 μg/mL puromycin for 1 week. The control cell line was generated following infection with a scrambled plasmid.

### 4.11. LDHA Overexpression

The plasmid pDEST27-LDHA was constructed by subcloning of LDHA cDNA (purchased from Korea Human Gene Bank, Daejeon, Korea) into pDEST27 (Invitrogen) vectors. Two sets of subcloned for LDHA were conducted. For reconstruction of the *LDHA*, the SNU620/5FU-shLDHA cells were transfected with pDEST27-LDHA and empty pDEST27 plasmid. Briefly, the cells were cultured up to 70% confluency. Then, the cells were treated with a mixture including 3 µg of DNA and Lipofectamine 2000 (Invitrogen) for 48 h. After incubation, the cells were selected with 200 μg/mL G418 for 1 week. Then, to confirm the efficacy of transfection, we performed the Western blot assay.

### 4.12. Apoptosis Analysis

Apoptotic cells were detected using the Annexin V-FITC Apoptosis Detection Kit (BD Biosciences, San Jose, CA, USA), according to the manufacture’s instruction. Briefly, cells were seeded at six-well plates (2 × 10^5^ cells/well) and treated with indicated concentrations of CA and 5FU for 2 days. After 2 days of treatments, the cell was washed with 1× PBS. The cells were suspended in 500 μL of binding buffer and treated with 5 µL of annexin V-FITC and 5 μL of propidium iodide (Sigma-Aldrich, St. Louis, MO, USA), followed by incubation for 15 min at room temperature in the dark. Fluorescence intensities were examined using a BD FACS CANTO II (BD Biosciences).

### 4.13. Mitochondrial Reactive Oxygen Species (ROS) Detection Assay

Mitochondrial ROS production was detected using a MitoSOX Red Mitochondrial Superoxide Indicator (Thermo Fisher Scientific). The cells were seeded at six-well plates (2 × 10^5^ cells/well) and Mito-TEMPO (20 μM; Sigma-Aldrich) was pretreated for 1 h before drug treatment. The cells were resuspended in 1 mL of 1× phosphate-buffered saline (PBS), after which 5 μM of MitoSOX was added. The cells were then incubated for 10 min at 37 °C. Fluorescence intensity was analyzed using a BD FACS CANTO II (BD Biosciences). Fluorescence image was detected by fluorescence microscope (magnification, 100×) (Axioimager M1 microscope, Carl Zeiss, Oberkochen, Germany).

### 4.14. DAPI Staining of Nucleus

SNU620/5FU was seeded in a 24-well plate (5 × 10^4^ cells/well) and treated with the indicated concentrations of CA and 5FU for 48 h. After washing with 1× PBS, cells were resuspended in 1-mL 1× PBS. The cells were then stained with 4 μg/mL DAPI for 30 min at room temperature and examined under a fluorescence microscope (magnification, 200×) (Carl Zeiss).

### 4.15. Statistical Analysis

The results of cell viability, lactate production, ECAR, OCR, qRT–PCR, LDHA, and LDHB activities, apoptosis, and mitochondrial ROS were indicated relative to control values and expressed as mean ± standard error of the mean of three independent experiments. Differences above the mean value of each group were analyzed by Student’s *t*-test, whereas differences between groups were analyzed by one-way analysis of variance with Tukey’s post hoc test using GraphPad Prism (Version 5.0, GraphPad Software, San Diego, CA, USA).

## 5. Conclusions

Taken together, resistant gastric cancer SNU620/5FU cells have glycolytic phenotypes, including elevated lactate production and higher LDHA expression than those in parental SNU620 cells. Restricting glycolysis with CA, as an LDHA-specific inhibitor, sensitizes SNU620/5FU cells to 5FU. Additionally, cotreatment with CA and 5FU increased mitochondrial ROS and apoptotic cell death in 5FU-resistant cells. Our findings suggest that CA may be a promising candidate for the development of an adjuvant drug that reduces resistance to 5FU-based chemotherapy by restricting LDHA activity.

## Figures and Tables

**Figure 1 ijms-22-05406-f001:**
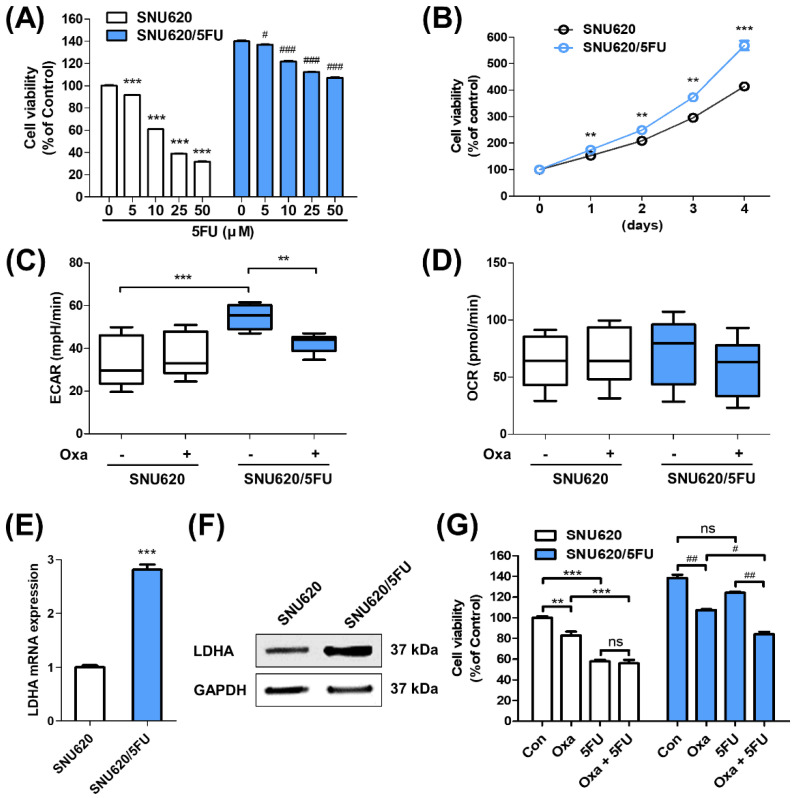
According to the results, 5-fluorouracil (5FU)-resistant cells have an elevated glycolytic characteristic: (**A**) SNU620 and SNU620/5FU cells were treated with the indicated concentrations of 5FU for 48 h. The viability of these cells was evaluated via MTT assay; (**B**) growth levels of SNU620 and SNU620/5FU cells were measured each day for 4 days with MTT assay; (**C**,**D**) SNU620 and SNU620/5FU cells were treated with oxamate (5 mM) or not for 24 h, and ECAR and OCR values were evaluated by Seahorse XF analyzer; (**E**,**F**) the expressions of LDHA in SNU620 and SNU620/5FU cells were detected via qRT–PCR and Western blot analysis; (**G**) SNU620 and SNU620/5FU cells were treated with oxamate (25 mM) and/or 5FU (10 μM) for 48 h. The results are shown as mean ± SEM. ** *p* < 0.01 and *** *p* < 0.001; # *p* < 0.05, ## *p* < 0.01, and ### *p* < 0.001, compared with the respective control. ECAR, extracellular acidification rates; OCR, oxygen consumption rates; LDHA, lactate dehydrogenase A. The experiments were independently performed in triplicate.

**Figure 2 ijms-22-05406-f002:**
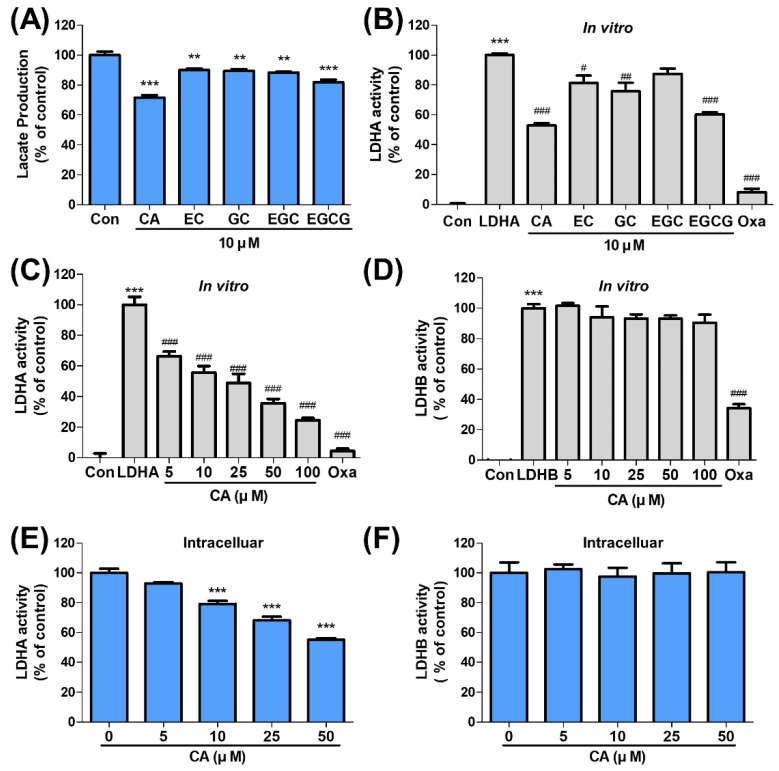
Catechin (CA) inhibits LDHA activity but not its expression: (**A**) lactate production of SNU620/5FU was measured following treatment with CA derivatives (10 μM) for 24 h; (**B**) in vitro LDHA activity was evaluated upon treatment with CA derivatives (10 μM) and oxamate (20 mM); (**C**,**D**) in vitro, LDHA and LDHB activities were estimated with the indicated concentrations of CA; (**E**,**F**) the cells were treated with indicated concentrations of CA for 24 h. Intracellular LDHA and LDHB activities were measured using the cell lysates as an enzyme source. The results are shown as mean ± SEM. ** *p* < 0.01 and *** *p* < 0.001, compared to the control (1st column). # *p* < 0.05, ## *p* < 0.01, and ### *p* < 0.001, compared to the negative control (second column). LDHA, lactate dehydrogenase A; LDHB, lactate dehydrogenase B. The experiments were independently performed in triplicate.

**Figure 3 ijms-22-05406-f003:**
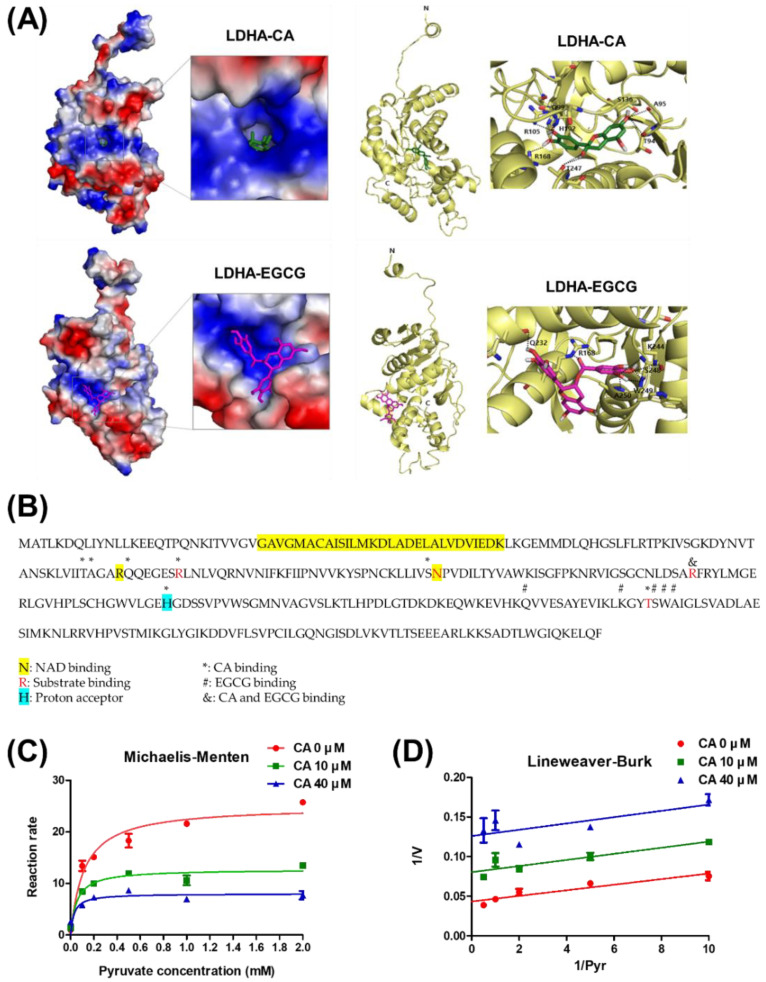
Mode of action of LDHA inhibition by catechin (CA): (**A**) surface representation (left panel) and ribbon diagram (right panel) of LDHA with CA/EGCG are shown; (**B**) sequence alignment of the CA and EGCG to LDHA; (**C**,**D**) LDHA enzyme kinetics was measured by LDHA activity assay using different doses of pyruvate (0.1, 0.5, 1, and 2 mM), and/or CA (0, 10, 40 μM). The fluorescence of NADH was examined with a spectrofluorometer. Michaelis–Menten curves and Lineweaver–Burk plots are shown to verify the inhibition mode of CA. The results are shown as mean ± SEM from three individual experiments.

**Figure 4 ijms-22-05406-f004:**
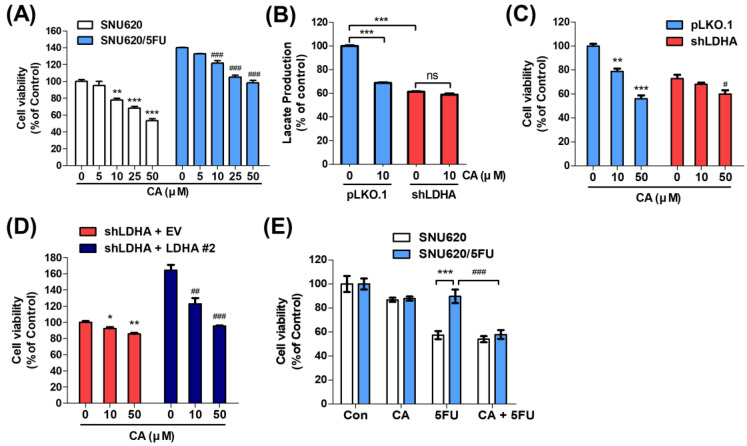
Cell viability was inhibited by cotreatment with catechin (CA) and 5-fluorouracil (5FU) in 5FU-resistant cells: (**A**) viabilities of SNU620 and SNU620/5FU cells were assessed following treatment with the indicated concentrations of CA for 48 h; (**B**) SNU620/5FU-pLKO.1 and SNU620/5FU-shLDHA cells were treated with CA (10 μM) for 24 h. Lactate production was measured in the culture media of SNU620/5FU-pLKO.1 and SNU620/5FU-shLDHA cells; (**C**,**D**) SNU620/5FU-pLKO.1, SNU620/5FU-shLDHA, SNU620/5FU-shLDHA+EV, and SNU620/5FU-shLDHA+LDHA#2 cells were treated with the indicated concentrations of CA for 48 h. The cell viability was measured using an MTT assay; (**E**) SNU620 and SNU620/5FU cells were treated with CA (10 μM) and/or 5FU (10 μM) for 48 h. The viabilities of these cells were measured via MTT assay. The results are shown as mean ± SEM. * *p* < 0.05, ** *p* < 0.01, and *** *p* < 0.001; # *p* < 0.05, ## *p* < 0.01, and ### *p* < 0.001, compared with the respective control. The experiments were independently performed in triplicate.

**Figure 5 ijms-22-05406-f005:**
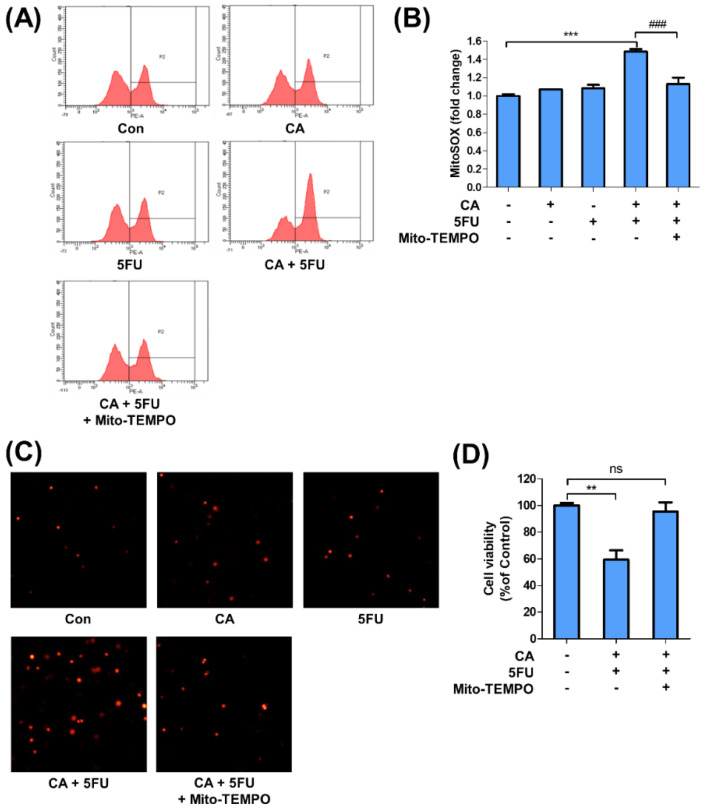
Cotreatment with catechin (CA) and 5-fluorouracil (5FU) increases the production of mitochondrial ROS: (**A**) SNU620/5FU cells were treated with the indicated concentrations of CA, 5FU (10 μM), and Mito-TEMPO (20 μM) for 48 h. The mitochondrial ROS of the cells was measured via FACS analysis using MitoSOX™ Red; (**B**) the bar graph shows the rate of cells that were positive for MitoSOX staining; (**C**) fluorescence microscopy images (X100) of SNU620/5FU cells stained with MitoSOX were presented; (**D**) the viabilities of the cells were measured via MTT assay. The results are shown as mean ± SEM. ** *p* < 0.01, *** *p* < 0.001, and ### *p* < 0.001, compared to the respective control. The experiments were independently performed in triplicate.

**Figure 6 ijms-22-05406-f006:**
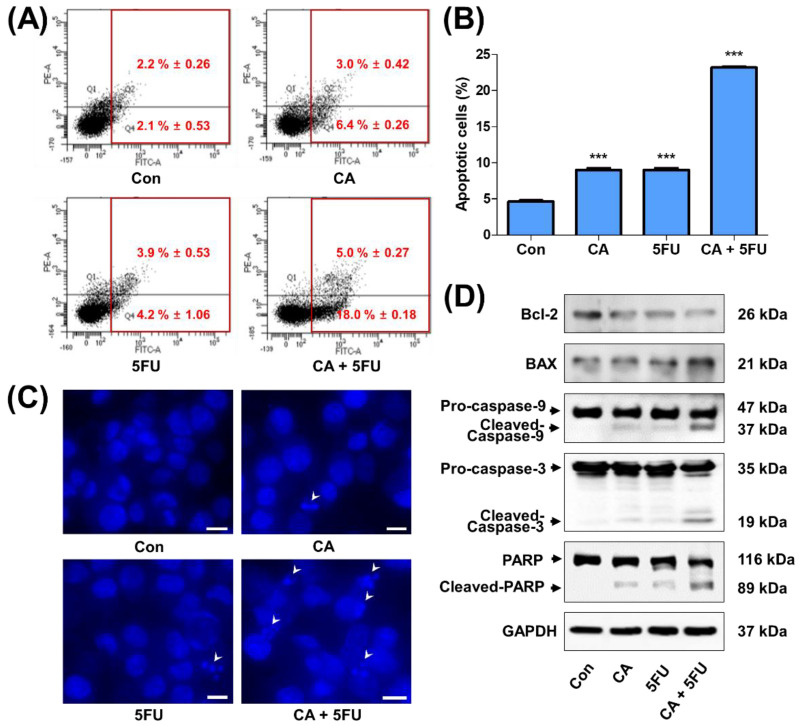
Increase of apoptosis by catechin (CA) and 5-fluorouracil (5FU) cotreatment; SNU620/5FU cells were treated with CA (10 μM) and/or 5FU (10 μM) for 48 h: (**A**) the number of apoptotic cells was analyzed via FACS analysis using PI-Annexin V staining; (**B**) the bar graph indicates the percentage of cells in early and late apoptotic phases. The results are shown as mean ± SEM. *** *p* < 0.001, compared to the control; (**C**) the nuclei of the cells were stained with DAPI, and fluorescence microscopic images were taken (X200). The white arrowhead indicates apoptotic cells. Scale bar, 10 μm; (**D**) the expression levels of proteins related to the apoptotic pathway were measured by Western blot analysis. The experiments were independently performed in triplicate.

**Figure 7 ijms-22-05406-f007:**
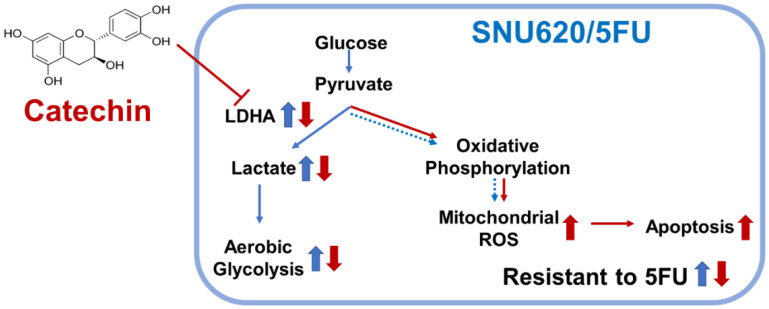
A schematic representation of the mechanism underlying the sensitizing effect of CA on 5FU-resistant cells. CA reduced the activity of LDHA, lactate production, and aerobic glycolysis, which were upregulated in SNU620/5FU. As a consequence of LDHA inhibition, increased mitochondrial ROS enhanced the apoptotic cell death and thereby reduced the resistance to 5FU. The increased pathways in SNU620/5FU cells are indicated by blue arrows, and actions of catechin are shown as red arrows.

**Table 1 ijms-22-05406-t001:** Primer sequences used for qRT–PCR.

Gene	Forward Primer	Reverse Primer
*LDHA*	5′-ACCGTGTTATTGGAAGCGGT-3′	5′-CTCCATGTTCCCCAAGGACC-3′
*PDK1*	5′-CTATGAAAATGCTAGGCGTCT-3′	5′-AACCACTTGTATTGGCTGTCC-3′
*PDK2*	5′-AGGACACCTACGGCGATGA-3′	5′-TGCCGATGTGTTTGGGATGG-3′
*PDK3*	5′-GCCAAAGCGCCAGACAAAC-3′	5′-CAACTGTCGCTCTCATTGAGT-3′
*PDK4*	5′-ACAGACAGGAAACCCAAGCC-3′	5′-CGATGTGAATTGGTTGGTCTGG-3′
*Rn18s*	5′-GTAACCCGTTGAACCCCATT-3′	5′-CCATCCAATCGGTAGTAGCG-3′

## Data Availability

The data will be made available upon reasonable request.

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
