# Peer review of "Targeting Lactate Dehydrogenase A with Catechin Resensitizes SNU620/5FU Gastric Cancer Cells to 5-Fluorouracil"

_ijms, 2021, doi:10.3390/ijms22105406_

Round 1

Reviewer 1 Report

The MS describes interesting work that suggests that CA inhibits LDHA activity and thereby, increases sensitivity to 5FU.

The work seems to be elaborate and detailed, many experiments were conducted using multidisciplinary tools and approaches, leading to the ability to conclude clear biological conclusions, based on in vitro and in silico research.

Comments:

The MS as it stands now is very hard to read.

  1. Many of the abbreviations are missing. Examples include: ECAR, OCR appear in line 84 but the abbreviations are brought only in section 4.4; ITC in line149, PDK in line 92, ETC in line 243, all are not defined in the text. Please see that all abbreviations are defined and explained when they first appear in text.

  2. Description of results is poor. for example, shLDHA is not describe throughout the MS, definitely not in section 2.4. Therefore it is very hard to interpret Fig 4. For example, once shLDHA is not defined and explained it is unclear why cell that express  shLDHA have lower level of lactate (Fig 4B)?  similarly, why cell viability of shLDHA expressing cells is lower (Fig 4D)?  what is LDHA #2? the effect of CA on cell viability in shLDHA cells and control seems similar, at least in concentration of 50 microM? These are just some examples out of many. Results should be re-written to clearly described each of the results, while defining all abbreviations when they first appear in text.

  3. Fig 6E is not referred to in the text. Perhaps it is Fig 5E? in any case- it should be separate from Fig 6 and presented as Fig 7 to my opinion. This Figure too, should be clearly explain. It is unclear what the arrows stand for? 5FU resistant or e.g., increase in apoptosis?

  4. Fig 1- A shows cell proliferation of SNU620/5FU, but not in G. if both where normalized to control, what might be the difference between the results?
  5. n (no of biological replicates) should be brought in all Fig legends.
  6. line 140- "may interact" and not "interacted" as this was only predicted based on modeling.
  7. deleted "surprisingly" from abstract- not a scientific term.

Reviewer 2 Report

It is apparent that manuscript entitled „Targeting Lactate Dehydrogenase A with Catechin Resensitizes SNU620/5FU Gastric Cancer Cells to 5-fluorouracil“ is written by experienced scientific group. It is useful and original study. However, before publication, several comments must be implemented.

Introduction:

  • PDHA1, PDKs are analyzed in Results, however, they are not mentioned in Introduction, please correct.
  • Last paragraph of Introduction represents Conclusion of the study! Delete it from this place. Conclusions are already inserted in the end of this study. Comprehensively describe study aims in this place.
  • Explain abbr. ECAR and OCR in the first mention in the text (not only in Methods).

Results:

  • L124, use “Gene expression levels of p-PDHA1,…”
  • L178-179, I do not agree with this sentence. Fig. 4E shows effectivity of CA+5FU also in SNU620 cells (decrease by 50% in cell viability vs CONT). Please, modify this sentence.
  • 6E use as a separate Figure, enlarge it and use explanations for arrows, including dashed lines.

Discussion:

  • I strongly suggest to develop ideas regarding combined anticancer therapy, because the aim of the preclinical studies and also this study must be the translation of experimental data to clinical practice. I strongly suggest to paraphrase sentences:

In aggressive malignant neoplasm, a highly variable sensitivity to therapeutics can be observed and some of cell lines can develop resistance to the treatment; the combined effects of bioactive compounds and conventional chemotherapy should be higher compared to a single compound, slowing the development of resistance. Therefore, the biological effects of specific phytochemicals with proven cytotoxic effects administered with conventional chemotherapy to target a wider range of signaling pathways in cancer cells including cancer metabolism and mitochondrial functions should be superior compared to single compound in cancer management and may delay the development of resistance.

Consider this excellent reference to cite this concept: Samuel et al. Biomolecules. 2019 Dec 9;9(12):846.

I congratulate authors to this valuable study!

Reviewer 3 Report

In this manuscript by Han et al., the authors have investigated the use of anti-glycolytic agents to overcome 5FU resistance. They found catechin as a potential drug candidate for the production of a new adjuvant that reduces chemoresistance to 5FU by limiting LDHA. The authors should address the below points before reconsidering for publication,

  1. An additional 5-FU cell line should be included to confirm the findings.
  2. Figure 2 can be moved to the supplementary section as it has only supporting data.? The rationale for studying the ROS levels should be clearly mentioned. 
  3. Fluorescence images of mitoSOX should be provided in Figure 5.
  4. In Figure 6, panel C, the authors should clearly explain the basis for confirming the apoptosis using DAPI images.
  5. In figure 6, panel D, better western blot images should be provided for Cleaved caspase-3 and cleaved PARP.

Round 2

Reviewer 1 Report

The MS is much improved now, results are better presented and explained.

Reviewer 3 Report

The manuscript was improved substantially by revision and can be accepted for publication.